# A population-based serological study of post-COVID syndrome prevalence and risk factors in children and adolescents

Roxane Dumont ⬤[1], Viviane Richard ⬤[1], Elsa Lorthe ⬤[1], Andrea Loizeau[1], Francesco Pennacchio ⬤[1], María-Eugenia Zaballa ⬤[1], Hélène Baysson[1,2], Mayssam Nehme ⬤[2], Anne Perrin ⬤[3], Arnaud G. L'Huillier[3,4], Laurent Kaiser ⬤[5,6,7], Rémy P. Barbe ⬤[8], Klara M. Posfay-Barbe ⬤[3,4], Silvia Stringhini ⬤[1,2,9,20], SEROCoV-KIDS study group* & Idris Guessous ⬤[2,10,20] ✉

Post-COVID syndrome remains poorly studied in children and adolescents. Here, we aimed to investigate the prevalence and risk factors of pediatric post-COVID in a population-based sample, stratifying by serological status. Children from the SEROCoV-KIDS cohort study (State of Geneva, Switzerland), aged 6 months to 17 years, were tested for anti-SARS-CoV-2 N antibodies (December 2021-February 2022) and parents filled in a questionnaire on persistent symptoms in their children (lasting over 12 weeks) compatible with post-COVID. Of 1034 children tested, 570 (55.1%) were seropositive. The sex- and age-adjusted prevalence of persistent symptoms among seropositive children was 9.1% (95%CI: 6.7;11.8) and 5.0% (95%CI: 3.0;7.1) among seronegatives, with an adjusted prevalence difference (ΔaPrev) of 4.1% (95%CI: 1.1;7.3). Stratifying per age group, only adolescents displayed a substantial risk of having post-COVID symptoms (ΔaPrev = 8.3%, 95%CI: 3.5;13.5). Identified risk factors for post-COVID syndrome were older age, having a lower socioeconomic status and suffering from chronic health conditions, especially asthma. Our findings show that a significant proportion of seropositive children, particularly adolescents, experienced persistent COVID symptoms. While there is a need for further investigations, growing evidence of pediatric post-COVID urges early screening and primary care management.

Evidence to date indicates that children, like adults, can experience post-COVID syndrome, also known as long COVID, with possibly major consequences on daily life[1,2]. The scientific community agreed on its definition in the pediatric population in March 2022[3]: "Post-COVID-19 condition occurs in young people with a history of confirmed SARS-CoV-2 infection, with at least one persisting physical symptom for a minimum duration of 12 weeks after initial testing that impacts everyday functioning and cannot be explained by an alternative diagnosis".

Many uncertainties remain regarding its prevalence, diagnosis, duration and treatment[4,5], partly due to clinical and methodological challenges. For example, there had been a lack of an official and standardized post-COVID definition in the pediatric population until very recently and a strong heterogeneity in study design and population across studies since the beginning of the pandemic[6,7]. In a recent systematic review[4] (22 studies from 12 countries including 23,141 children and adolescents, of which five studies had population-based control groups), the prevalence of post-COVID symptoms (lasting more than

---

A full list of affiliations appears at the end of the paper. *A list of authors and their affiliations appears at the end of the paper.
✉e-mail: Idris.Guessous@hcuge.ch

3 months) was 2–8% higher in the seropositive than in the control group, with a higher prevalence difference in adolescents. This prevalence was lower than suggested by other studies that did not use control groups[5], which may be due to several explanations. First, many children might experience long-lasting symptoms due to other viruses or medical events or due to the general stressful pandemic environment. Therefore, differentiating symptoms linked specifically to a SARS-CoV-2 infection from other diagnoses using a control group of non-infected persons is paramount to avoid overestimation of post-COVID syndrome prevalence. Second, most studies on post-COVID syndrome rely on samples of confirmed infection (RT-PCR and antigen tests)[4,8] hence excluding asymptomatic cases and underestimating the proportion of infected children and adolescents as this population was not systematically tested. Serological data allow to precisely estimate the proportion of infected children and adolescents by including asymptomatic and mild cases. Accordingly, the difference of prevalence between seropositives and seronegatives can provide an accurate estimator of post-COVID syndrome prevalence and/or unexpected complications such as acute hepatitis in the general pediatric population. The most frequently declared pediatric post-COVID symptoms are fatigue, headache, shortness of breath, chronic cough and myalgia with a higher risk among girls and adolescents[5]. Adults also develop similar symptoms, including persistent cough, fever, headache, chest pain, hair loss, loss of taste and smell, amongst many others[9].

In addition to the uncertainties previously mentioned, only very few studies have so far analysed risk factors of pediatric post-COVID syndrome. A large body of literature has shown the adverse effects of low socioeconomic conditions on several health outcomes[10]. Similar mechanisms could be expected in the pediatric post-COVID population.

In this study, we aimed to assess the prevalence of persistent symptoms lasting over 12 weeks after a SARS-CoV-2 infection comparing seropositive children and adolescents with their seronegative counterparts, using a representative sample of the general population of the canton of Geneva. We also aimed to identify risk factors for experiencing persistent symptoms.

## Results
### Descriptive results
Among 3060 households, 625 households participated in our study (participation rate of 20.4%) (Fig. 1).

Our sample included 1034 children aged 6 months to 17 years from 612 households: 505 (49%) were girls and the mean age was 10.2 years [SD = 4.2]. Overall, 785 (76%) referent parents had a tertiary education, 200 (19%) a secondary education and 42 (4%) had a primary education level (seven missing data). Among our participants, 150 (15%) were reported to live in a household with an average to poor financial situation; and 270 (26%) were reported to have a chronic medical condition (Table 1, S1). There were 570 (55%) children who tested positive for anti-SARS-CoV-2 N antibodies. Overall, we observed that 253 (24%) had a documented confirmed COVID-19 infection, and 198 (19%) declared an episode of acute symptoms that could be considered as a symptomatic SARS-CoV-2 infection. Among the 79 (8%) children who had experienced persistent COVID-19-related symptoms lasting over 12 weeks since the beginning of the pandemic, 54 (68%) were seropositive and 25 (32%) were seronegative.

Compared to seronegatives, seropositive children tended to experience more often symptoms such as abdominal pain, trouble concentrating, smell loss, dripping nose, muscle pain, breathing difficulties, headache and constipation. In contrast, seronegative children were declared as having more anxiety, lower mood and dermatological symptoms (skin rash). Only abdominal pain, smell loss and constipation were significantly more common in seropositive subjects (Fig. 2, Table S4). The severity of symptoms lasting over 12 weeks was on average slightly higher among seropositive children (mean [sd] = 4.9 [2.5]) compared to seronegative subjects (mean [sd] = 4.2 [2.5]),

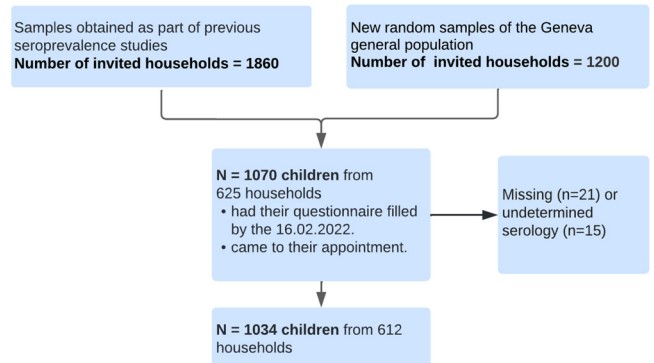

**Fig. 1 | Participants recruitment and inclusion into analytical sample.** The Flowchart illustrates the process of recruitment and participation.

although not statistically significant (*p*-value = 0.1) (Table 1). Among the 54 (9.5%) seropositive children who suffered from symptoms lasting over 12 weeks (Table S2), 31 (57%) were adolescents aged 12–17 years. A confirmed infection was documented in 30/54 (56%) and 26/54 (48%) reported acute symptoms at the time of infection.

### Adjusted prevalence of persistent symptoms overall and stratified by age
The adjusted prevalence of persistent symptoms among seropositive and seronegative children was 9.1% (95%CI: 6.7;11.8) and 5.0% (95%CI: 3.0;7.1), respectively. The corresponding adjusted prevalence difference (ΔaPrev) was 4.1% (95%CI: 1.1;7.3). After stratification by age groups, the prevalence of post-COVID syndrome was higher among adolescents (8.3%, 95%CI: 3.5;13.5) than among younger children (0.0%, 95%CI: −5.2;5.2 among 6–11 years old and 4.2%; 95%CI: −4.4;13.3 among 0–5 years old) (Table 2), in which no difference was observed. After stratification by sex, the prevalence within females was of 3.4% (95%CI: 1.1;8.4) and in males, of 4.7% (95%CI: 0.2;9.4).

### Risk factors for persistent symptoms
Sex- and age-adjusted prevalence ratios were estimated for symptoms lasting over 12 weeks. Older age (aPR 1.1, 95%CI: 1.0;1.2, continuous variable), being seropositive (aPR 1.8, 95%CI: 1.2;2.8), suffering from a chronic condition (aPR 3.6, 95%CI: 2.3;5.5) and living in a household with an average to poor financial situation (aPR 2.5, 95%CI: 1.4;4.6) were risk factors for experiencing persistent symptoms. Sex was not associated with long lasting symptoms (aPR 1.1, 95%CI: 0.8;1.6).

In a sub-analysis restricted to seropositive participants (*N* = 570), we also observed that the risk factors associated with persistent symptoms were older age (aPR 1.1, 95%CI: 1.0;1.2), suffering from a chronic condition (aPR 3.5, 95%CI: 2.0;6.1) and living in a household with an average to poor financial situation (aPR 3.0, 95%CI: 1.5;6.2). Lower level of parental education was related to a higher risk of persistent symptoms, although not statistically significant (Table 3). Based on the adjusted difference of frequency (%) of chronic health conditions within children experiencing 12 weeks persistent symptoms (*n* = 40), we observed a significant positive difference between seropositives and seronegatives only with asthma (Table S3).

## Discussion
Our results showed that a significant proportion of children and adolescents suffered from persistent symptoms compatible with post-COVID syndrome. The overall estimated prevalence in the pediatric population was about 4%. Stratifying per age group, only adolescents displayed a substantial risk of having post-COVID symptoms. Risk factors for post-COVID syndrome were older age, having a lower socioeconomic status and suffering from chronic health conditions, especially asthma.

## Table 1 | Descriptive statistics

| Characteristics | Overall, n = 1034 | Seronegative, n = 464 | Seropositive, n = 570 | p-value[a] |
|---|---|---|---|---|
| Sex | | | | 0.07 |
| Female | 528 (51%) | 252 (54%) | 276 (48%) | |
| Male | 505 (49%) | 212 (46%) | 293 (51%) | |
| Other | 1 (0%) | 0 (0%) | 1 (0%) | |
| Age group (years) | | | | 0.02 |
| 0–5 | 160 (16%) | 80 (17%) | 80 (14%) | |
| 6–11 | 445 (43%) | 178 (38%) | 267 (47%) | |
| 12–17 | 429 (41%) | 206 (44%) | 223 (39%) | |
| Parental education[b,c] | | | | 0.09 |
| Tertiary | 785 (76%) | 366 (79%) | 419 (74%) | |
| Secondary | 200 (19%) | 79 (17%) | 121 (21%) | |
| Primary | 42 (4%) | 15 (3%) | 27 (5%) | |
| Financial situation[b,c] | | | | 0.40 |
| Good | 822 (79%) | 376 (81%) | 446 (79%) | |
| Average to poor | 150 (14%) | 59 (13%) | 91 (16%) | |
| Do not want to answer | 60 (6%) | 27 (4%) | 33 (5%) | |
| Chronic condition[d] | 270 (26%) | 133 (29%) | 137 (24%) | 0.09 |
| Confirmed SARS-CoV-2 infection[e,f] | 253 (24%) | 25 (5%)[f] | 228 (40%) | <0.001 |
| Confirmed SARS-CoV-2 symptomatic[g] infection | 198 (19%) | 21 (4%) | 177 (31%) | <0.001 |
| Vaccination Status[h,i] | | | | <0.001 |
| No | 432 (75%) | 211 (71%) | 221 (79%) | |
| Yes, 1 dose | 31 (5%) | 4 (1%) | 27 (10%) | |
| Yes, 2 doses | 116 (20%) | 83 (28%) | 33 (12%) | |
| Persistent symptoms | | | | |
| Symptoms lasting over 4 weeks | 175 (17%) | 73 (16%) | 102 (18%) | 0.50 |
| Symptoms lasting 4–6 weeks | 57 (5%) | 27 (6%) | 30 (5%) | 0.70 |
| Symptoms lasting 6–8 weeks | 27 (3%) | 13 (3%) | 14 (2%) | 0.70 |
| Symptoms lasting 8–12 weeks | 12 (1%) | 8 (2%) | 4 (1%) | 0.13 |
| Symptoms lasting over 12 weeks | 79 (8%) | 25 (5%) | 54 (9%) | 0.01 |
| Median number[j] | | | | |
| Symptoms lasting over 4 weeks | 2 (1–4) | 2 (1–4) | 2 (1–4) | 0.40 |
| Symptoms lasting over 12 weeks | 2 (1–4) | 2 (1–2) | 2 (1–5) | 0.20 |
| Symptoms severity (1–10 scale)[k] | | | | |
| Symptoms lasting over 4 weeks | 4.6 (2.5) | 4.2 (2.5) | 4.9 (2.5) | 0.10 |
| Symptoms lasting over 12 weeks | 4.7 (2.6) | 4.2 (2.6) | 4.9 (2.6) | 0.20 |

[a]Two-sided Fisher's exact test or Pearson's Chi-squared test.
[b]Missing data (Parental education (n = 7), Household financial situation (n = 2)).
[c]Answers are based on the referent parent.
[d]Medical/physical condition diagnosed by a health professional that lasted (or was expected to last) longer than 6 months.
[e]Diagnosed COVID-19 infection with a positive test (RT-PCR or rapid antigen detection test). Positive SARS-CoV-2 test (RT-PCR or antigen tests) reported by parents.
[f]The difference between serological results and confirmed SARS-CoV-2 infection could be explained by test performance and errors in answering the questionnaire.
[g]Parent could also declare whether the child(ren) had experience an episode of acute symptoms that could be considered as a symptomatic SARS-CoV-2 infection.
[h]Vaccination for children aged 5 to 11 opened the 04.01.22, many children were not eligible at the time of the recruitment.
[i]Data stratified by age group are presented in Table S1.
[j]Median (Interquartile range Q1-Q3).
[k]Mean (sd).

We defined post-COVID syndrome in children based on the recent definition published by Stephenson et al. in March 2022[3]. The estimated overall prevalence of post-COVID syndrome in our sample is high, yet lower than in other studies[4]. There might be multiple reasons. First, designs, samples and characteristics of post-COVID varied considerably across studies. Second, our data included mildly symptomatic and asymptomatic children and, therefore, a greater number of infected children than when relying on confirmed infection alone as testing was never systematic in children. Third, we worked with a population-based sample rather than clinical registries. Finally, a proportion of seropositive children in our sample might have been infected too recently making it impossible to studied long term persistent symptoms.

Our data showed a meaningful difference in prevalence between children >12 years and younger individuals, in agreement with the study by Stephenson et al.[5] We could expect this proportion to be higher in reality as parents might not be aware of all of their children's symptoms; adolescents may be less open about their issues since they are in a period of their life seeking more independence.

The prevalence difference between seronegatives and seropositives (~8%) in adolescents is worth investigating as it potentially represents a very high absolute number, although lower than in adults[11]. Post-COVID syndrome might worsen adolescents' lives, already negatively impacted by the general pandemic environment[12,13] and could exacerbate long-term negative consequences on health, social and academic outcomes. On the contrary, we did not observe significant differences between seronegative and seropositive children <12 years[14]. The lack of difference in reported persistent symptoms in children <12 years could be partly explained by the relatively small sample size and by symptoms often developed after common respiratory infections other than SARS-CoV-2 in this age range.

The pandemic-related difficulties to meet psychological and physical developmental needs may have been exacerbated by persistent symptoms, further preventing children from going about their daily activities and worsening their mental health. Most declared persistent symptoms among seropositives were abdominal pain, trouble concentrating, smell loss, dripping nose, muscle pain, breathing difficulties, headache and constipation, in agreement with the recent literature[4,15,16]. The severity of symptoms reported by parents was, on average, slightly higher in seropositive children, suggesting that persistent symptoms following a SARS-CoV-2 infection might be more severe and represent a more substantial burden on their daily life[14,17]. Sher et al. underlined that depressive symptoms appear to be common in patients with post-COVID syndrome. Long-term medical and psychological consequences of post-COVID syndrome could drastically worsen mental health and even increase suicidal ideation and behaviors[18]. Similar outcomes could be expected in the pediatric population, especially in adolescents. Post-COVID syndrome should not be underestimated in children and calls for appropriate medical strategies including reinforcing vaccination strategies[19].

In an analysis restricted to seropositives, children from households with lower socioeconomic status were more likely to experience post-COVID syndrome. This is in line with the growing literature on health inequalities of the COVID-19 pandemic in terms of incidence, testing and severity of the infection[20]. Those inequalities could be explained by differential exposure to the virus, greater susceptibility to infection, stronger comorbidities in vulnerable groups associated with severe outcomes, and disparities in healthcare. Similar mechanisms could explain the greater exposure of vulnerable children to the post-COVID syndrome. Also, it is reasonable to highlight that COVID-19 vaccination uptake was more frequent among individuals with higher socioeconomic status and is considered as the major protection against an acute or severe SARS-CoV-2 infection, hence developing a post-COVID syndrome[21].

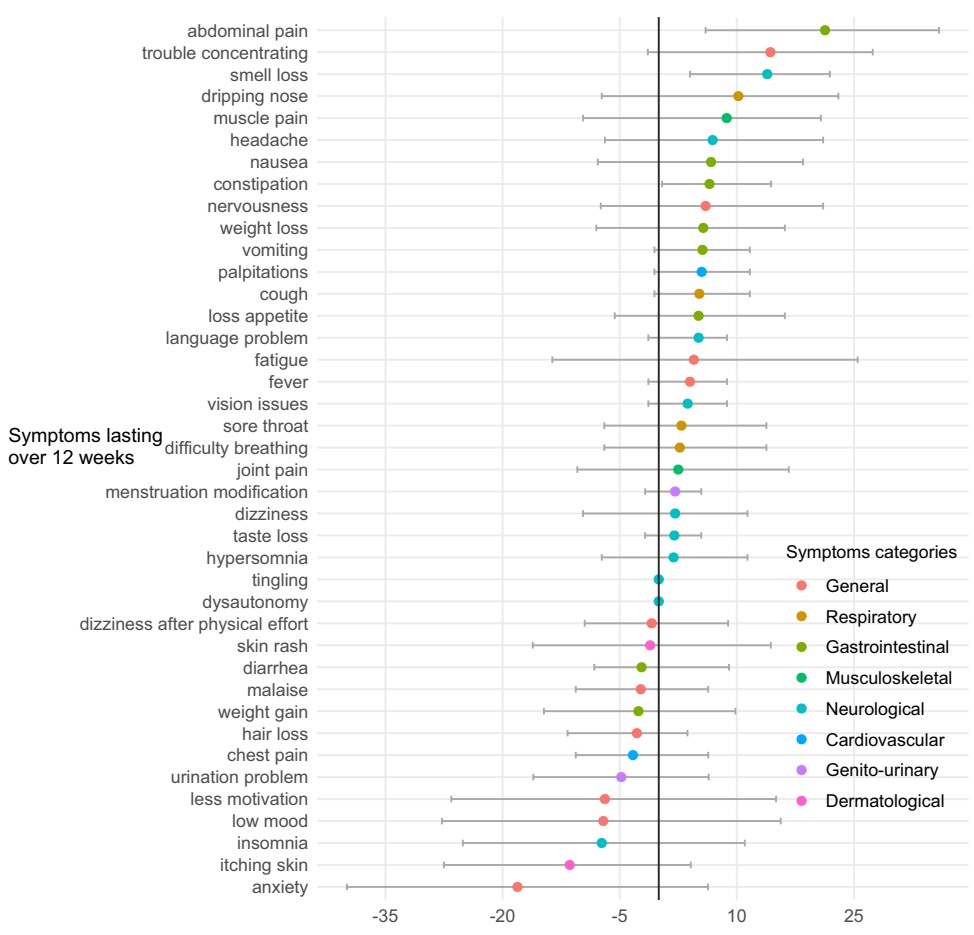

**Fig. 2 | Sex- and age- adjusted difference (%) of symptoms lasting over 12 weeks between seropositive and seronegative children, December- February 2022, Geneva, Switzerland.** Sample size: 79 children and adolescents whom experienced symptoms lasting over 12 week. The colored dots refer to types of symptoms, as presented in the legend. Data are presented as mean values of the differences between seropositive and seronegative children with +/−Standard Error of the Mean (SEM), illustrated by the error bars.

In our sample, post-COVID seems more prevalent among children with chronic disease, especially asthma, although causality could be bi-directional. In general, asthmatic children have a higher risk of respiratory infections. However, the association between asthma and SARS-CoV-2 infection remains unclear in the pediatric population, as many large-scale ecological studies presented reduced pediatric asthma during the pandemic likely due to physical distancing, masks, and perhaps decreases in air pollution. On the other hand, other research focused on the individual level and presented asthma as a risk factor for hospitalization in children with COVID-19, but not for worse COVID-19 outcomes[22,23].

The major strength of our study is the population-based design with a wide age range, covering from babies to adolescents. Only very few studies on post-COVID syndrome rely on random population samples and include children of this age range[4]. We identified previous SARS-CoV-2 infections relying on serological data, which also detects previous asymptomatic and mild infection. Unlike studies relying on test-confirmed infections, which suffer from selection bias, relying on serological assessment yields a better estimation of the proportion of the infected population.

However, the use of serological data also poses challenges. In particular, it is impossible to determine the exact date of infection only relying on this data[5]. Our estimates are therefore based on a comparison of children with and without anti-SARS-CoV-2 anti-bodies, using a control group to avoid overestimating the pre-valence of post-COVID symptoms. Although the diagnosis of post-

COVID condition related to persistent symptoms could not be medically assessed by excluding all other alternative diagnoses, the comparison between seropositive and seronegative children enabled to control for the occurrence of persistent symptoms unrelated to a SARS-CoV-2 infection. Once again, this highlights the importance of a control group and the complexity of identifying the post-COVID syndrome methodologically and clinically. Further-more, despite the fact that some parents were probably aware of their children's exposure status when answering the questionnaire due to confirmed COVID-19 diagnosis or a symptomatic infection, relying on serological tests mitigates the bias of parents over-reporting persistent symptoms when knowing their child(ren) had been infected[24].

Our study also has several limitations. Data were parent-reported and their answers could be influenced by their own experience or the household environment. Also, they might not be aware of some of their children's symptoms, particularly for adolescents. Furthermore, as the time of infection and the date of onset persistent symptoms were unknown in many cases, we are not able to study the incident risk of post-COVID, for which another study design including the time of the infection and a follow-up would be necessary. Moreover, we can-not exclude that some children might suffer from long-lasting symp-toms due to Omicron infection but were not identified as such in our analysis as they had not reached the 12-week threshold that we used to define persistent symptoms. This could have led to an underestimation of the prevalence of post-COVID.

**Table 2 | Sex and age- adjusted prevalence and prevalence difference of persistent symptoms**

| Age group (years) | Serological status | Prevalence of symptoms lasting over 12 weeks Percent[a,b,c] (95%CI) | Prevalence difference of symptoms lasting over 12 weeks Percent[a,b,c] (95%CI) |
|---|---|---|---|
| Female (n = 528)[d] | | | |
| | Negative | 4.9 (2.2;7.6) | |
| | Positive | 8.3 (4.9;11.6) | |
| | **Difference** | | 3.4 (−1.1;8.4) |
| Male (n = 505)[d] | | | |
| | Negative | 5.0 (2.12;8.0) | |
| | Positive | 9.7 (6.4;13.2) | |
| | **Difference** | | 4.7 (0.2;9.4) |
| 0–5 (n = 160) | | | |
| | Negative | 3.8 (0.0;8.1) | |
| | Positive | 8.0 (1.8;14.2) | |
| | **Difference** | | 4.2 (−4.4;13.3) |
| 6–11 (n = 445) | | | |
| | Negative | 5.3 (2.0;8.6) | |
| | Positive | 5.3 (2.6;8.1) | |
| | **Difference** | | 0.0 (−5.2;5.2) |
| 12–17 (n = 429) | | | |
| | Negative | 5.3 (2.3;8.4) | |
| | Positive | 13.6 (9.3;18.1) | |
| | **Difference** | | 8.3 (3.5;13.5) |
| All ages (n = 1034) | | | |
| | Negative | 5.0 (3.0;7.1) | |
| | Positive | 9.1 (6.7;11.8) | |
| | **Difference** | | 4.1 (1.1;7.3) |

[a]Adjusted for age and sex, when appropriate.
[b]A model additionally adjusting for chronic condition, financial situation and parental education yielded similar results.
[c]Unadjusted estimates are presented in Table S5.
[d]As presented in Table 1, one child had no biological sex assigned.

Despite the high number of randomly recruited children from a wide age range, the participation rate was relatively low, although in line with participations rates in children cohort in other countries[25]. In addition to the general challenge of recruiting children in population-based cohort[26], we faced the additional difficulty that parents were often reluctant to have their children, especially young ones, undergo a blood draw for research purposes only. We also noticed that some parents were tired of hearing about the pandemic, which also represented a strong barrier to recruitment. Finally, individuals with favorable socio-economic conditions were more likely to participate. This could have led to an underestimation of the prevalence of post-COVID since their occurrence was more common among underprivileged individuals. Overall, this might limit the representativeness of our results.

It would be interesting to study persistent symptoms in children and adolescents who were more recently affected with the Omicron variant as less virulent variants could potentially lead to lower prevalence of post-COVID.

Our study contributes to a better understanding of post-COVID in the pediatric population. The impact of COVID-19 vaccination could

not be assessed and further studies should explore if prior vaccination reduces the risk of developing post-COVID.

A significant proportion of children experienced post-COVID symptoms lasting over 12 weeks after infection with an estimated prevalence of 4% overall and 8% in adolescents. Older age, having a chronic condition and living on a household with lower socioeconomic conditions were identified as risk factors for the post-COVID syndrome. Our understanding of post-COVID syndrome will likely evolve as scientific evidence grows. Nevertheless, it is fundamental to rapidly implement effective primary care management, including early screening and detection, and health promotion to assist children and adolescents suffering from this syndrome who might experience long term physical and mental adverse consequences.

## Methods

### Study design and data collection
Data are drawn from the SEROCoV-KIDS study, an ongoing, longitudinal and prospective cohort study, which aims at monitoring and evaluating direct and indirect impacts of the COVID-19 pandemic on the health and development of children and adolescents.

Children and adolescents' eligibility criteria included (1) being between 6 months and 17 years old, (2) residing in the Canton of Geneva at the time of enrollment, and (3) either being newly selected from random samples obtained from state registries or having a household member already participating in a population-based COVID-19 seroprevalence study conducted by our group[27–29]. These population-based samples were provided by the Swiss Federal Statistical Office (FSO).

At the baseline assessment, all children were invited to perform a serological test (by blood drawing) to measure anti-SARS-CoV-2 antibodies (anti-N). One of the parent or legal guardian (referent parent) was asked to fill out online questionnaires (see supplementary material) related to health and development for him/herself and for each of his/her children, on the Specchio-COVID19 secured digital platform[30].

The Geneva Cantonal Commission for Research Ethics approved this study (ID 2021-01973). Referent parents of participants, as well as adolescents aged 14 years or older, provided written informed consent. Younger children provided oral consent.

The baseline assessment consisted of a blood sample for serological testing and online questionnaires. Non-respondents received email reminders 4, 7 and 21 days after the first questionnaire invitation and up to two phone calls were made.

The data were collected using the softwares Formstack® (02.2022) and sugarCRM® (version 4.2).

### Study population
The baseline assessment of the SEROCoV-KIDS cohort took place from December 1st, 2021 to April 30th, 2022. We decided to restrict our analyses by excluding post-COVID syndrome after Omicron infection. Considering that post-COVID is by definition symptoms persisting at least 3 months after the infection, and that Omicron variants (BA.1/BA.2) became dominant in Switzerland at the end of December 2021, with a surge of COVID-19 in children in January 2022, we included exclusively children recruited between December 1st, 2021 and February 16th, 2022. These participants include those reporting post-COVID symptoms who would have had their infection prior to these dates, and thus prior to the Omicron surge. Children for whom we failed to take a blood sample were excluded from this analysis.

### Symptoms evaluation
At baseline, parents were systematically asked if their child(ren) had suffered from symptoms lasting at least 4 weeks since the beginning of the pandemic, no matter their SARS-CoV-2 infectious status (Supplement). The duration of symptoms was then classified as

**Table 3 | Adjusted prevalence ratio of symptoms lasting over 12 weeks**

| Report of symptoms lasting over 12 weeks | | Overall[a] n = 1034[b] PR (95%CI) | p-value | Within seropositives[a] n = 570[b] PR (95%CI) | p-value | Within seronegatives[a] n = 464[b] PR (95%CI) | p-value |
|---|---|---|---|---|---|---|---|
| Sex | Female | 1.0 (ref) | – | 1.0 (ref) | – | 1.0 (ref) | |
| | Male | 1.1 (0.8;1.6) | 0.665 | 1.1 (0.7;1.8) | 0.723 | 0.9 (0.4–1.9) | 0.875 |
| Age (years) | | 1.1 (1.0;1.2)* | 0.012 | 1.1 (1.0;1.3)* | 0.021 | 1.1 (0.9–1.2) | 0.261 |
| Serological status | Seronegative | 1.0 (ref) | – | – | – | – | – |
| | Seropositive | 1.8 (1.2;2.8)** | <0.01 | – | – | – | – |
| Chronic condition | No | 1.0 (ref) | – | 1.0 (ref) | – | 1.0 (ref) | – |
| | Yes | 3.6 (2.3;5.5)**c,d | <0.01 | 3.5 (2.0;6.1)**c,d | <0.01 | 2.9 (1.3–6.5)**c,d | <0.01 |
| Parental education | Tertiary | 1.0 (ref) | – | 1.0 (ref) | – | 1.0 (ref) | – |
| | Secondary | 1.2 (0.7;2.0)c,d | 0.471 | 1.3 (0.6;2.5)c,d | 0.469 | 1.2 (0.7;2.0)c,d | 0.469 |
| | Primary | 1.9 (0.8;4.7)c,d | 0.365 | 1.7 (0.5;5.2)c,d | 0.369 | 1.9 (0.8;4.6)c,d | 0.369 |
| Financial situation | High | 1.0 (ref) | – | 1.0 (ref) | – | 1.0 (ref) | – |
| | Average to poor | 2.5 (1.4;4.6)*c,d | <0.05 | 3.0 (1.5;6.2)*c,d | <0.05 | 1.2 (0.9–3.4)c,d | 0.501 |

[a]Prevalence ratio and 95% confidence interval are from Poisson regression with random effect on the household using the GLMMadaptive package in R and are adjusted for age, sex, or both according to independent variable, using a two-sided Likelihood Ratio Test, without adjustments for multiple comparisons.
[b]Complete case analysis. For each model presented, missing data in the covariates were excluded.
[c]Adjusting for age and sex.
[d]Unadjusted estimates are presented in Table S6.
*indicates p-value < 0.05.
**indicates p-value < 0.01.

lasting 4–6 weeks, 6–8 weeks, 8–12 weeks or >12 weeks. Parents had to report the approximate date of the start of persistent symptoms. They could select symptoms from an exhaustive list of COVID-19-related symptoms based on a literature review at the time of the questionnaire design and revised by a group of expert physicians on post-COVID. The persistent symptoms were grouped into seven general categories: general, respiratory, gastrointestinal, cardiovascular, musculoskeletal, neurological and dermatological symptoms. The severity of persistent symptoms was then evaluated by asking parents "considering the most severe symptom of the episode, to what extent did this symptom affect the child's daily life (on a scale of 1 very weak limitation–10 strong limitation)". Parents were not aware of their child(ren) serological test results while answering the questionnaire.

### SARS-CoV-2 infection

Confirmed SARS-CoV-2 infection was defined from a positive SARS-CoV-2 test (RT-PCR or antigen tests) reported by parents, along with the date of the first positive test. Parent could also declare whether the child(ren) had experience an episode of acute symptoms that could be considered as a symptomatic SARS-CoV-2 infection (confirmed symptomatic infection). Serological tests were based on the semi-quantitative commercially-available immunoassay Roche Elecsys anti-SARS-CoV-2 N, detecting total Ig (including IgG) against the nucleocapsid protein of the SARS-CoV-2 virus. Seropositivity was defined using the manufacturer's cut-off of index ≥ 1.0[31]. The test has an in-house sensitivity of 99.8% (95% CrI, 99.4%–100%) and specificity of 99.1% (95% CI, 98.3–99.7%) (Roche Diagnostics, Rotkreuz, Switzerland).

Roche Anti-N assays maintained high sensitivity over time and were identified as appropriate tool for screening infection[31] and anti-N antibodies have been shown to also persist in time in pediatric populations[32,33].

The antibodies detected by this test (anti-N antibodies) are produced following an infection but not following vaccination with mRNA vaccines. At the time of the study, the two authorized vaccines in Switzerland were the mRNA-1273 from Moderna/US NIAID[34] and the mRNA-BNT162b2/Comirnaty-[35] from Pfizer/BioNTech6. Both elicit a response exclusively to the S protein of SARS-CoV-2, as opposed to natural infections, which typically elicit a response to both the N and S virus proteins.

### Other characteristics

Other variables were described, including sociodemographic characteristics such as age, sex, parental education and household financial status. Parental education (based on the referent parent) was categorized into three groups: primary education (compulsory schooling), secondary education (apprenticeship and high school) and tertiary education (universitary studies). Household financial status was defined as "average to poor" if the referent parent chose one of the following statements about their financial situation: "I have to be careful with my expenses and an unexpected event could put me into financial difficulty" or "I cannot cover my needs with my income and I need external support". If the referent parent had selected the following statement "I am comfortably off, money is not a concern and it is easy for me to save" or "My income covers my expenses and covers any minor contingencies" about their household financial situation was defined as "good". Referent parents were also asked if their child(ren) suffered from a chronic health condition, defined as a medical/physical condition diagnosed by a health professional that lasted (or was expected to last) >6 months.

### Statistical analysis

We compared sociodemographic and health-related characteristics between children who tested positive for anti-SARS-CoV-2 N antibodies and children who tested negative, overall and stratified into three age groups (0–5, 6–11 and 12–17 years old) using Chi-squared and Student-t tests, as appropriate.

We used marginal prediction after logistic regression to estimate prevalence and prevalence difference, adjusting for age and sex. This statistical approach corresponds to models in which conditional predicted probabilities are calculated for each exposure level with every confounder fixed at its mean value. Mixed-effect Poisson regression with robust variance, based on the sandwich estimator[36], was used to estimate prevalence ratio[37] and correct for potential dependence between participants as some children were siblings. Participants with missing data in at least one of the covariates (n = 9, 1%) were excluded from the models. Statistical significance was defined as a level of confidence of 95% and all analyses were performed with R (version 4.0.3), using GLMMadaptive (0.8-5), dplyr (1.0.10), gtsummary (1.6.2) packages.

### Reporting summary

Further information on research design is available in the Nature Portfolio Reporting Summary linked to this article.

## Data availability

Participants' informed consent did not authorize data to be immediately publicly available. It does allow, however, for the data to be made available to the scientific community upon submission of a data request application to the investigators board via the corresponding author. All requests for data are responded within 3 months from submission.

## Code availability

Our computer code are accessible to researchers upon request to the corresponding author.

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

## Acknowledgements

We thank the Jacobs foundation and the Federal Office of Public Health (FOPH) for supporting the project. We are grateful to the staff of the Unit of Population Epidemiology of the HUG Division of Primary Care Medicine as well as to all participants whose contributions were invaluable to

the study. We also acknowledge all the members of the SEROCoV-KIDS study group.

## Author contributions

The idea for the study originally came from I.G., S.S., K.M.P.-B., R.P.B., H.B., E.L., A.L., V.R. and R.D. R.D., E.L. and H.B. carried out the literature search. S.S., I.G., H.B., F.P., M.-E.Z. and V.R. conceptualized, designed the study and M.-E.Z., F.P., A.L. and V.R. acquired the data. Members of the SEROCoV-KIDS study group took part in the data collection process. R.D. performed data analysis. R.D., E.L. and I.G. wrote the first draft of the manuscript. A.P., A.G.L., K.M.P.-B., L.K. and M.N. contributed to the interpretation of data and critically reviewed the manuscript for important intellectual content. All authors have read, critically revised and approved the final version of this manuscript.

## Competing interests

The authors declare no competing interests.

## Additional information

[1]Unit of Population Epidemiology, Division of Primary Care Medicine, Geneva University Hospitals, Geneva, Switzerland. [2]Department of Health and Community Medicine, Faculty of Medicine, University of Geneva, Geneva, Switzerland. [3]Division of General Pediatrics, Department of Woman, Child, and Adolescent Medicine, Geneva University Hospitals, Geneva, Switzerland. [4]Pediatric Infectious Diseases Specialist, Geneva University Hospitals and Faculty of Medicine, Geneva, Switzerland. [5]Geneva Center for Emerging Viral Diseases and Laboratory Virology, Geneva University Hospitals, Geneva, Switzerland. [6]Division of Laboratory Medicine, Geneva University Hospitals, Geneva, Switzerland. [7]Department of Medicine, Faculty of Medicine, University of Geneva, Geneva, Switzerland. [8]Division of Child and Adolescent Psychiatry, Department of Woman, Child, and Adolescent Medicine, Geneva University Hospitals, Geneva, Switzerland. [9]University Center for General Medicine and Public Health, University of Lausanne, Lausanne, Switzerland. [10]Division and Department of Primary Care Medicine, Geneva University Hospitals, Geneva, Switzerland. [20]These authors contributed equally: Silvia Stringhini, Idris Guessous. ✉e-mail: Idris.Guessous@hcuge.ch

## SEROCoV-KIDS study group

Roxane Dumont [1], Deborah Amrein[1], Andrew S. Azman[1,11], Antoine Bal[1,12], Michael Balavoine[13], Rémy P. Barbe [8], Hélène Baysson[1,2], Julie Berthelot[1], Patrick Bleich[1], Livia Boehm[1], Gaëlle Bryand[1], Viola Bucolli[1], Prune Collombet[1], Alain Cudet[14], Vladimir Davidovic[1], Carlos de Mestral Vargas[1], Paola D'Ippolito[1], Richard Dubos[1], Isabella Eckerle[5,6], Marion Favier[13], Nacira El Merjani[1], Natalie Francioli[1], Clément Graindorge[1], Séverine Harnal[1], Samia Hurst[2], Laurent Kaiser [5,6,7], Omar Kherad[15], Julien Lamour[1], Pierre Lescuyer[16], Arnaud G. L'Huillier[3,4], Andrea Loizeau[1], Elsa Lorthe [1], Chantal Martinez[1], Stéphanie Mermet[17], Mayssam Nehme [2], Natacha Noël[1], Francesco Pennacchio [1], Javier Perez-Saez[1,11], Anne Perrin [3], Didier Pittet[18], Jane Portier[10], Klara M. Posfay-Barbe [3,4], Géraldine Poulain[16], Caroline Pugin[1], Nick Pullen[1], Viviane Richard [1], Frederic Rinaldi[14], Jessica Rizzo[1], Deborah Rochat[1], Cyril Sahyoun[17], Irine Sakvarelidze[1], Khadija Samir[1], Hugo Alejandro Santa Ramirez[1], Stephanie Schrempft[1], Claire Semaani[1], Stéphanie Testini[1], Yvain Tisserand[7,19], Deborah Urrutia Rivas[1], Charlotte Verolet[1], Jennifer Villers[1], Guillemette Violot[12], Nicolas Vuilleumier[7,16], Sabine Yerly[6], María-Eugenia Zaballa [1], Christina Zavlanou[19], Silvia Stringhini [1,2,9,20] & Idris Guessous [2,10,20]✉

[11]Department of Epidemiology, Johns Hopkins Bloomberg School of Public Health, Baltimore, MD, USA. [12]Communication Directorate, Geneva University Hospitals, Geneva, Switzerland. [13]Médecine et Hygiène, Chem. de la Mousse 46, 1225 Chêne-Bourg Geneva, Switzerland. [14]DotBase SA, Rte des Acacias 25 CH – 1227 Carouge, Geneva, Switzerland. [15]Division of Internal Medicine, Hôpital de la Tour and Faculty of Medicine, Geneva, Switzerland. [16]Departments of Diagnostic, Geneva University Hospitals, Geneva, Switzerland. [17]Pediatric Reception and Emergency Service, Geneva University Hospitals, Geneva, Switzerland. [18]Infection Control Program and World Health Organization Collaborating Center on Patient Safety, Geneva University Hospitals and Faculty of Medicine, Geneva, Switzerland. [19]Swiss Center for Affective Sciences, University of Geneva, Geneva, Switzerland.

