## [Peer Review File · Nature Communications]

A population-based serological study of post-COVID syndrome prevalence and risk factors in children and adolescentsREVIEWER COMMENTS

Reviewer #1 (Remarks to the Author):

The report presents data on long COVID in children. Seropositive vs seronegative children were studied, through questionnaires filled in by the parents. The cohort is population-based, and not taken from clinical records, which makes this study particularly interesting for the calculations of rates of longCOVID in pediatrics.

The main finding reported by the authors is that risk factors for post-COVID syndrome in pediatrics are older age, having a lower socioeconomic status and suffering from chronic health conditions, especially asthma.

The document is overall very well written. The English writing is good, and at the journal standard, I believe. References are appropriate and the limitations section well delineated. Discussion, conclusion and claims are fair and grounded.

This review has been conducted according to:

- STROBE checklist
- CASP checklist
- COPE guidelines

STRENGTHS of this manuscript:

- The topic is of potential wide interest.
- The potential impact on the healthcare system and public health is high
- The study design is correct.

WEAKNESS

- Data are not open and code is only available upon request.
- The study does not control or take into account for SARS-CoV-2 variants. Both incidence of post-COVID and symptoms experienced might in principle change in dependence of SARS-CoV-2 variants. This aspect is not tackled by the study.
- Results are no longer new; however, these are very good confirmatory results.
- I understand statistical thresholds are exploratory, and multiple comparisons are not corrected for. This is probably due to the sample size, which is not huge.
- The cohort is probably slightly underpowered for these analyses

MAJOR COMMENTS:

I do not have any major comment.

MINOR COMMENTS:

- Line 67: I am not entirely sure "syndrome is a plural. So "Post-COVID syndrome remain" might not be correct and need an 's'. Please check grammar.
- Line 107 "partly due to clinical and methodological challenges. Please, provide a couple of examples with "such as..."
- Line 141: please, add a comma in "17 years old 2) residing in...", which then becomes "17 years old, 2) residing in..."
- Line 158-159: Is there any specific reason for selecting this timeframe for inclusion? (i.e., dec 2021 to feb 2022)
- Line 160: How many children were lost at the stage of blood sample collection?
- Line 170 "The severity of the persistent symptoms was then evaluated for the persistent symptoms" this sentence can probably be simplified, and repetitions avoided.
- Line 186-187: "the only ones approved to date for children in Switzerland." This specification is totally unclear to me. Could the authors please clarify here?
- Line 247: "in which no difference was observe." Should be "in which no difference was observed." Please amend.
- Line 342-343: " our recruitment took place from December 2021, until February 2022, a period during which many children got infected (Omicron wave)." Can the authors explain this point please? To my understanding, children recruited in this period might well have been infected by wild, alpha or delta variants. If infected by omicron, conversely, they might not have had sufficient timeframe to develop long COVID, which requires 12 weeks by definition to be assessed. If the

authors mean something different, this probably needs to be much better clarified.
- Please extend caption for figure 1. Caption should be self-explanatory.

Reviewer #2 (Remarks to the Author):

This is an interesting study that further highlights the impact of PCC in children. The study is overall well written and designed, several limitations intrinsic to the study design has been mentioned and acknowledged.

Although I support this paper, I think the major issue the authors have to clarify is how they calculated symptoms duration, since most of their job was based on serological data without (from what I understand) a known starting infection in the large majority of cases. This need to be clarified before moving the paper forward

INTRO

- please add reference toward the appropriateness of using serology as a screening of infection in children, including its persistence over time. This should be specifically referred to the anti N serology
- several important papers on the topic have not included in the intro nor discussion, specifically important papers by Munblit et al and Buonsenso et al

METHODS

- the authors should provide in the supplementary material the questionnaire they proposed to families, and should also clarify if this was a self filled questionnaire or if symptoms were collected by a doctor during a real medical evaluation.
- also, the authors should provide info about how they might have excluded alternative diagnosis, which is part of the PCC definition provided by both the Clock Delphi and the WHO. If this was not done (as I think), this should be mentioned in the limitations, being a serious study limitation.
- it is unclear how the authors wanted to collect the info that symptoms began with the infection. How the authors could assess this? I mean a PCC symptom should be considered as a new symptom never reported before the infection
- it is unclear to me how the authors considered the >12weeks persisting symptoms in those children with unknown previous sars-cov-2 infection, or in the control group. this needs further clarification. did the authors did this analyses only on a subgroup of patients with a known starting day of a specific symptom?

RESULTS:

well presented, although I have some doubts about how the authors could calculate the duration of symptoms in those that do not recall an initial infection

- can the authors analyze somehow the impact of a previous vaccination on symptoms persistence?
- please add a supplementary table with all collected symptoms in the two groups. I am curious, for example, to see how much specific symptoms like anosmia (dysgeusia) were reported in the seronegative group

DISCUSSION

- one of the reasons that can explain lack of differences of reported symptoms in younger children can be related by the overall high frequency of symptoms like cough, upper airway involvement, rashes, eg common symptoms that develop after common respiratory infections, which have been circulating a lot after society returned to mostly normal routines. This concept should be mentioned as a possible explanation of the lack of differences in young children, along with a real possibility of an age-gradient risk of long covid with increasing age
- the discussion should include more literature on recent advances in the field of long covid in children, including recent studies by Buonsenso et al team which showed objective evidence of lung perfusion in a child, brain PET abnormalities, long term olfactory dysfunction, immune dysregulation, and two other studies from Germany showing abnormalities of lung function documented by lung MRI. Such data are necessary to put in context the results of your study, as most of the mentioned symptoms in the >12yro group are being studied by other centers.

Reviewer #3 (Remarks to the Author):

Thank you for the opportunity to review this manuscript on an important topic; long-COVID in children and adolescents. I enjoyed reading it, and I hope my comments are helpful. This is a combined sero- and questionnaire survey looking at the association between SARS-CoV-2 infection and persisting symptoms in 1034 Swiss children and adolescents (Geneva canton, 570 sero-positives).

The primary strength of the study is the inclusion of serological data on participants. The primary weakness is the lack of information on when infection happened in the seropositive participants. Parents might well report on symptoms that started before infection (they can report on symptoms from since the start of the pandemic).

The study reports a significant difference in the prevalence of persistent symptoms among sero-positives compared to sero-negatives.

- 1) Please specify when baseline is in this study?
- 2) In the figure it states that all questionnaires were filled out by February 16 2022. This would mean that omicron infections could not contribute to persistent symptoms >12 wks. However, I assume that omicron infection could be responsible for sero-positives?
- 3) You state that serological data assures that parents are blinded to exposure status. What about the positive PCR-test results? Also, I would expect that the parents would know about any symptomatic infections among their children; at least they would suspect that the child had been infected. Please comment?
- 4) Participation rate was on the low side (20%). Please comment?
- 5) Lines 208-209: Please provide more details on what is meant by "marginal prediction after logistic regression".
- 6) Table 1: Please clarify if "Confirmed SARS-CoV-2 symptomatic infection" is based on the parental recall information that is mentioned in lines 177-179.
- 7) Why do you only adjust for age and sex, and not more of the variables from table 1 such as chronic condition, financial situation and parental education.
- 8) Please provide 1) the median number of symptoms reported among test-positives and test-negatives, respectively, 2) the median number of symptoms lasting over 12 weeks among test-positives and test-negatives, respectively.
- 9) How is the mean of the severity score calculated; I assume, that each child can contribute with more than one persistent symptom.
- 10) Please consider presenting the mean of the severity score by specific symptom and by sero-pos./sero.neg.
- 11) Table 2: Please also provide prev.diff. according to sex.
- 12) Table 3: Why not present these associations within test-negatives instead of within all? It is not clear to me that these "risk factors" are unique to test-positives. Why not estimate a ratio of ratios between test-positives and test-negatives to see if there is a difference.
- 13) Table S1: Please replace "ans" in the column headings.

Manuscript reference: NCOMMS-22-36030

Title: Post-COVID syndrome prevalence and risk factors in children and adolescents: A population-based serological study

Authors: Roxane Dumont, Viviane Richard, Elsa Lorthe, Andrea Loizeau, Francesco Pennacchio, María-Eugenia Zaballa, H el ene Baysson, Mayssam Nehme, Anne Perrin, Arnaud G. L'Huillier, Laurent Kaiser, R emy P. Barbe, Klara M. Posfay-Barbe, Silvia Stringhini, Idris Guessous & SEROCov-KIDS study group

Page and line numbers refer to the version of the manuscript with revisions.

Comments from Reviewer #1

Comment 1:

The report presents data on long COVID in children. Seropositive vs seronegative children were studied, through questionnaires filled in by the parents. The cohort is population-based, and not taken from clinical records, which makes this study particularly interesting for the calculations of rates of longCOVID in pediatrics. The main finding reported by the authors is that risk factors for post-COVID syndrome in pediatrics are older age, having a lower socioeconomic status and suffering from chronic health conditions, especially asthma. The document is overall very well written. The English writing is good, and at the journal standard, I believe. References are appropriate and the limitations section well delineated. Discussion, conclusion and claims are fair and grounded.

This review has been conducted according to:

- STROBE checklist
- CASP checklist
- COPE guidelines

STRENGTHS of this manuscript:

- The topic is of potential wide interest.
- The potential impact on the healthcare system and public health is high
- The study design is correct.

WEAKNESS

- Data are not open and code is only available upon request.
- The study does not control or take into account for SARS-CoV-2 variants. Both incidence of post-COVID and symptoms experienced might in principle change in dependence of SARS-CoV-2 variants. This aspect is not tackled by the study.
- Results are no longer new; however, these are very good confirmatory results.
- I understand statistical thresholds are exploratory, and multiple comparisons are not corrected for. This is probably due to the sample size, which is not huge.
- The cohort is probably slightly underpowered for these analyses

Response: We thank the reviewer for their careful reading of the manuscript and their constructive remarks. We are also grateful for highlighting the strengths and weaknesses of our study.

MAJOR COMMENTS:

I do not have any major comment.

MINOR COMMENTS:

- 1) Line 67: I am not entirely sure “syndrome is a plural. So “Post-COVID syndrome remain” might not be correct and need an ‘s’. Please check grammar.

Response: We thank the reviewer for noticing this error, we corrected the sentence.

- 2) Line 107 “partly due to clinical and methodological challenges. Please, provide a couple of examples with “such as...”

Response: As suggested by the reviewer, we added the following details, with a recent reference from *Munblit et al.*

“Many uncertainties remain regarding its prevalence, diagnosis, duration and treatment, partly due to clinical and methodological challenges. For example, there had been a lack of an official and standardized post-COVID definition in the pediatric population until very recently and a strong heterogeneity in study design and population across studies since the beginning of the pandemic [7]”. *Lines 107-110*

[7] Munblit D, Buonsenso D, Sigfrid L, Vijverberg SJ, Brackel CL. Post-COVID-19 condition in children: a COS is urgently needed. *Lancet Respir Med.* 2022;10(7):628–629.

- 3) Line 141: please, add a comma in “17 years old 2) residing in...”, which then becomes “17 years old, 2) residing in...”.

Response: The change has been made.

- 4) Line 158-159: Is there any specific reason for selecting this timeframe for inclusion? (i.e., dec 2021 to feb 2022)

Response: We thank the reviewer for raising this point. We added this section in the revised manuscript in order to clarify the timeframe of the study.

“The baseline assessment of the SEROCov-KIDS cohort took place from December 1st, 2021 to April 30th, 2022. We decided to restrict our analyses by excluding post-COVID syndrome after Omicron infection. Considering that post-COVID is by definition symptoms persisting at least three months after the infection, and that Omicron variants (BA.1/BA.2) became dominant in Switzerland at the end of December 2021, with a surge of COVID-19 in children in January 2022, we included exclusively children recruited between December 1st, 2021 and February 16th, 2022. These participants include those reporting post-COVID symptoms whom would have had their infection prior to these dates, and thus prior to the Omicron surge.” *Lines 165-173*

- 5) Line 160: How many children were lost at the stage of blood sample collection?

Response: As illustrated in the flowchart, 36 children were excluded for missing or indeterminate serology. Among them, 21 were classified as failed blood draw and 15 were excluded because of indeterminate serology results.

These details have been added in the flowchart (Figure 1).

- 6) Line 170 “The severity of the persistent symptoms was then evaluated for the persistent symptoms” this sentence can probably be simplified, and repetitions avoided.

Response: The sentence has been simplified.

“The severity of persistent symptoms was then evaluated by asking parents “considering the most severe symptom of the episode, to what extent did this symptom affect the child’s daily life (on a scale of 1 very weak limitation - 10 strong limitation)” (On a scale from 1: very low limitation to 10: strong limitation).” *Lines 184-188*

- 7) Line 186-187: “the only ones approved to date for children in Switzerland.” This specification is totally unclear to me. Could the authors please clarify here?

Response: We apologize that the sentence was not clear, and have now modified this section as follows:

“The antibodies detected by this test (anti-N antibodies) are produced following an infection but not following vaccination with mRNA vaccines. At the time of the study, the two authorized vaccines in Switzerland were the mRNA-1273 from Moderna/US NIAID [8] and the mRNA-BNT162b2/Comirnaty [9] from Pfizer/BioNTech6. Both elicit a response exclusively to the S protein of SARS-CoV-2, as opposed to natural infections, which typically elicit a response to both the N and S virus proteins.” *Lines 205-210*

[8] Baden LR, El Sahly HM, Essink B, et al. Efficacy and Safety of the mRNA-1273 SARS-CoV-2 Vaccine. *N Engl J Med* 2020; 384: 403–16.

[9] Polack FP, Thomas SJ, Kitchin N, et al. Safety and Efficacy of the BNT162b2 mRNA Covid-19 Vaccine. *N Engl J Med* 2020; 383: 2603–15

- 8) Line 247: “in which no difference was observe.” Should be “in which no difference was observed.” Please amend.

Response: The sentence was corrected, thank you.

- 9) Line 342-343: “Our recruitment took place from December 2021, until February 2022, a period during which many children got infected (Omicron wave).” Can the authors explain this point please? To my understanding, children recruited in this period might well have been infected by wild, alpha or delta variants. If infected by omicron, conversely, they might not have had sufficient timeframe to develop long COVID, which requires 12 weeks by definition to be assessed. If the authors mean something different, this probably needs to be much better clarified.

Response: The reviewer is correct and this is why we constrained the study analysis to February 16th 2022, as presented in the response from comment #4.

The majority of children in our sample were seropositive due to wild, alpha- or delta-variants. However, as highlighted in the limitation section of our article and correctly mentioned by the reviewer, it is still possible that some children included in our sample were seropositive due to an infection in the early stages of the Omicron wave.

We hope we clarified this point with our revisions from comment #4. In addition, we completed a previous paragraph in the limitation section:

“Moreover, we cannot exclude that some children might suffer from long-lasting symptoms due to Omicron infection but were not identified as such in our analysis as they had not reached the 12-week threshold that we used to define persistent symptoms. This could have led to an underestimation of the prevalence of post-COVID.” *Lines 374-377*

- 10) Please extend caption for figure 1. Caption should be self-explanatory.

Response: Details were added to the caption of Figure 1 as follows:

“Participants recruitment and inclusion into analytical sample”

Comments from Reviewer #2

This is an interesting study that further highlights the impact of PCC in children. The study is overall well written and designed, several limitations intrinsic to the study design has been mentioned and acknowledged.

- 1) Although I support this paper, I think the major issue the authors have to clarify is how they calculated symptoms duration, since most of their job was based on serological data without (from what I understand) a known starting infection in the large majority of cases. This needs to be clarified before moving the paper forward.

Response: We thank the reviewer for this comment and careful review.

We assessed persistent symptoms at the population level by asking parents to report new symptoms that appeared after the start of the pandemic, and lasting for more than 12 weeks, independently of the child's infection or serological status. The 12 weeks threshold was defined based on the definition of post-COVID syndrome and was used as the duration of symptoms (more than 12 weeks). We then calculated the prevalence of post-COVID syndrome by calculating the difference between seropositive and seronegative, using seronegative individuals as controls for non-COVID related symptoms with the aim to yield an accurate estimation of post-COVID in children and correct for symptoms due to alternative diagnoses.

We acknowledge that knowing when the child was infected using PCR or antigen test data remains highly interesting when studying the temporality and characteristics of persistent symptoms due to COVID-19. However, as also reported in other population-based studies, children were not systematically tested throughout the pandemic and SARS-COV-2 infection in children is frequently asymptomatic, thereby making estimates based on confirmed cases biased by an inaccurate estimation of the number of infected children. Our study design aims to overcome this limitation using serological data as it allows a more precise estimation of the denominator.

To clarify how symptoms duration information were collected and how post-COVID was defined in this study, we changed the manuscript as follows:

“In this study, we aimed to assess the prevalence of persistent symptoms lasting more than 12 weeks after a SARS-CoV-2 infection comparing seropositive children and adolescents with their seronegative counterparts, using a representative sample of the general population of the canton of Geneva. We also aimed to identify risk factors for experiencing persistent symptoms.” *Lines 136-140*

INTRODUCTION

- 2) Please add reference toward the appropriateness of using serology as a screening of infection in children, including its persistence over time. This should be specifically referred to the anti N serology

Response: We thank the reviewer for this comment. Our group conducted a study on the persistence of anti-SARS-CoV-2 antibodies and immunoassay heterogeneity. Within a sample of adults, authors showed that antibodies were detected up to 9 months after the initial infection and that Roche Anti-N assays maintained a high sensitivity over time and were an appropriate tool for screening for infection [15]. Although not based on Roche-

N test, similar mechanisms have been observed in studies on anti-N persistence in children [16]. The reference is available in the manuscript, line 198. The subsequent sentence was added to clarify this point:

“Roche Anti-N assays maintained a high sensitivity over time and were identified as an appropriate tool for screening for infection [15]. Additionally, anti-N antibodies have been shown to persist in time in pediatric populations [16,17]” *Lines 200-202.*

[15] Perez-Saez J, Zaballa ME, Yerly S, Andrey DO, Meyer B, Eckerle I, et al. Persistence of anti-SARS-CoV-2 antibodies: immunoassay heterogeneity and implications for serosurveillance. *Clin Microbiol Infect.* 2021;27(11):1695–e7.

[16] Ireland, G., Jeffery-Smith, A., Zambon, M., Hoschler, K., Harris, R., Poh, J., ... & Ladhani, S. N. (2021). Antibody persistence and neutralising activity in primary school students and staff: prospective active surveillance, June to December 2020, England. *EClinicalMedicine*, 41, 101150.

[17] Wachter F, Regensburger AP, Peter AS, Knieling F, Wagner AL, Simon D, et al. Continuous monitoring of SARS-CoV-2 seroprevalence in children using residual blood samples from routine clinical chemistry. *Clin Chem Lab Med CCLM.* 2022;60(6):941–951.

- 3) Several important papers on the topic have not included in the intro nor discussion, specifically important papers by Munblit et al and Buonsenso et al-

Response: We thank the reviewer for pointing out those two interesting and recent studies on the subject. The mentioned papers have been added in the introduction.

Buonsenso D, Munblit D, De Rose C, Sinatti D, Ricchiuto A, Carfi A, et al. Preliminary evidence on long COVID in children. *Acta Paediatr Oslo Nor* 1992. 2021;110(7):2208. *Line 100*

Munblit D, Buonsenso D, Sigfrid L, Vijverberg SJ, Brackel CL. Post-COVID-19 condition in children: a COS is urgently needed. *Lancet Respir Med.* 2022;10(7):628–629. *Line 110*

METHODS

- 4) The authors should provide in the supplementary material the questionnaire they proposed to families, and should also clarify if this was a self-filled questionnaire or if symptoms were collected by a doctor during a real medical evaluation.

Response: We agree that this is an important aspect of the study. As mentioned in the methods, children’s symptoms are parent-reported (lines 153-157).

The questions used to collect our study data are now added in the supplementary material, translated from French to English.

- 5) Also, the authors should provide info about how they might have excluded alternative diagnosis, which is part of the PCC definition provided by both the Clock Delphi and the WHO. If this was not done (as I think), this should be mentioned in the limitations, being a serious study limitation.

Response: We thank the reviewer for this interesting point. In order to exclude alternative diagnoses, we used the prevalence difference between seronegatives and seropositives. Yet, because it is parent-reported, we cannot exclude residual misclassification.

We added this point in the limitation as follows:

“Although the diagnosis of post-COVID condition related to persistent symptoms could not be medically assessed by excluding all other alternative diagnoses, the comparison between seropositive and seronegative children enabled to control for the occurrence of persistent symptoms unrelated to a SARS-CoV-2 infection.” *Lines 359-363*

- 6) It is unclear how the authors wanted to collect the info that symptoms began with the infection. How the authors could assess this? I mean a PCC symptom should be considered as a new symptom never reported before the infection.

Response: We thank the reviewer for this comment.

Our study aimed to study persistent symptoms at the population level with a design that allowed to estimate the proportion of potential post-COVID in the population.

We first identified persistent symptoms by asking parents whether the child had experienced new symptoms that appeared after the start of the pandemic, and lasting over 12 weeks, no matter the child's SARS-CoV-2 infectious status. We then, stratified their answers by serological status. We were therefore able to identify which of the reported persistent symptoms could actually be considered as post-COVID comparing seropositive and seronegative individuals and calculating the sex- and age-adjusted difference between the two groups.

- 7) It is unclear to me how the authors considered the >12weeks persisting symptoms in those children with unknown previous sars-cov-2 infection, or in the control group. This needs further clarification. Did the authors did this analyses only on a subgroup of patients with a known starting day of a specific symptom?

Response: We thank the reviewer for raising this point. As it is closely related, please refer to the answers to comments #1 and #6. We hope this will clarify the issue.

RESULTS

- 8) Well presented, although I have some doubts about how the authors could calculate the duration of symptoms in those that do not recall an initial infection.

We thank you for this comment. The duration of symptoms as mentioned above was defined as lasting more than 12 weeks (definition of post-COVID condition), considering only new symptoms that appeared after the start of the pandemic. We hope this answer as well as the answers to comments #1, 6 and 7 could clarify this point.

- 9) Can the authors analyse somehow the impact of a previous vaccination on symptoms persistence?

Response: We thank the reviewer for this very interesting comment. We believe that it would be of great interest to evaluate the impact of a previous vaccination on symptoms persistence. However, information on vaccination status is not precise enough in our data. “Our study contributes to a better understanding of post-COVID in the pediatric population. The impact of COVID-19 vaccination could not be assessed and further studies should explore if prior vaccination reduces the risk of developing post-COVID”
Lines 394-397

- 10) Please add a supplementary table with all collected symptoms in the two groups. I am curious, for example, to see how much specific symptoms like anosmia (disgeusia were reported in the seronegative group)

Response: We thank the reviewer for their interest. We added Table S4, in which a detailed report of each symptom lasting over 12 weeks is presented.

In addition, Figure 2 was revised, as we noticed that the symptoms presented in Figure 2 were referring to symptoms lasting over 4 weeks and not 12 weeks. We also corrected the following sentences accordingly.

“Compared to seronegatives, seropositive children tended to experience more often symptoms such as abdominal pain, trouble concentrating, smell loss, dripping nose, muscle pain, breathing difficulties, headache and constipation. In contrast, seronegative children were reported to have more anxiety, lower mood and dermatological symptoms (skin rash). Only abdominal pain, smell loss and constipation were significantly more common in seropositive subjects (Figure 2, Table S4).” *Lines 254-259*

“Most frequently declared persistent symptoms among seropositives were abdominal pain, trouble concentrating, smell loss, dripping nose, muscle pain, breathing difficulties, headache and constipation, in agreement with a recent literature review” *Lines 320-322*

DISCUSSION

- 11) None of the reasons that can explain lack of differences of reported symptoms in younger children can be related by the overall high frequency of symptoms like cough, upper airway involvement, rashes, eg common symptoms that develop after common respiratory infections, which have been circulating a lot after society returned to mostly normal routines. This concept should be mentioned as a possible explanation of the lack of differences in young children, along with a real possibility of an age-gradient risk of long covid with increasing age

Response: We appreciate the reviewer's suggestion. We believe that the fact that we compare seronegative to seropositive children allows us to account for the comparable circulation of other respiratory viruses in the two groups.

- 12) The discussion should include more literature on recent advances in the field of long covid in children, including recent studies by Buonsenso et al team which showed objective evidence of lung perfusion in a child, brain PET abnormalities, long term olfactory dysfunction, immune dysregulation, and two other studies from Germany showing abnormalities of lung function documented by lung MRI. Such data are necessary to put in context the results of your study, as most of the mentioned symptoms in the >12yro group are being studied by other centers.

Response: We thank the reviewer for this comment. As suggested, we have added recent additional references in the discussion in order to contextualize our results. In addition to the two references added in the introduction above, we consolidated our literature review of the burden of persistent of the post-COVID condition by adding the following references.

Buonsenso D, Di Giuda D, Sigfrid L, Pizzuto DA, Di Sante G, De Rose C, et al. Evidence of lung perfusion defects and ongoing inflammation in an adolescent with post-acute sequelae of SARS-CoV-2 infection. *Lancet Child Adolesc Health*. 2021;5(9):677–680. *Line 322*

Heiss R, Wagner A, Tan L, Schmidt S, Regensburger AP, Ewert F, et al. Persisting pulmonary dysfunction in pediatric post-acute Covid-19. *medRxiv*. 2022 *Line 322*

Rao S, Lee GM, Razzaghi H, Lorman V, Mejias A, Pajor NM, et al. Clinical features and burden of postacute sequelae of SARS-CoV-2 infection in children and adolescents. *JAMA Pediatr*. 2022 *Line 325*

Comments from Reviewer #3

Thank you for the opportunity to review this manuscript on an important topic; long-COVID in children and adolescents. I enjoyed reading it, and I hope my comments are helpful. This is a combined sero- and questionnaire survey looking at the association between SARS-CoV-2 infection and persisting symptoms in 1034 Swiss children and adolescents (Geneva canton, 570 sero-positives). The primary strength of the study is the inclusion of serological data on participants. The primary weakness is the lack of information on when infection happened in the seropositive participants. Parents might well report on symptoms that started before infection (they can report on symptoms from since the start of the pandemic). The study reports a significant difference in the prevalence of persistent symptoms among sero-positives compared to sero-negatives.

- 1) Please specify when baseline is in this study?

Response: We thank the reviewer for raising this point. Also in response to comment #4 from Reviewer #1, we have added the following paragraph:

“The baseline assessment of the SEROCOVID-KIDS cohort took place from December 1st, 2021 to April 30th, 2022. We decided to restrict our analyses by excluding post-COVID syndrome after Omicron infection. Considering that post-COVID is by definition symptoms persisting at least three months after the infection, and that Omicron variants (BA.1/BA.2) became dominant in Switzerland at the end of December 2021, with a surge of COVID-19 in children in January 2022, we included exclusively children recruited

between December 1st, 2021 and February 16th, 2022. These participants include those reporting post-COVID symptoms whom would have had their infection prior to these dates, and thus prior to the Omicron surge.” *Lines 165-173*

- 2) In the figure it states that all questionnaires were filled out by February 16, 2022. This would mean that omicron infections could not contribute to persistent symptoms >12 wks. However, I assume that omicron infection could be responsible for sero-positives?

Response: The reviewer is correct, it is likely that some children in our sample were infected by Omicron but with an infection too recent to attribute potential persistent symptoms to that infection. In line with comment #9 from reviewer #1, we modified the text as follows:

“Moreover, we cannot exclude that some children might suffer from long-lasting symptoms due to Omicron infection but were not identified as such in our analysis as they had not reached the 12-week threshold that we used to define persistent symptoms. This could have led to an underestimation of the prevalence of post-COVID.” *Lines 374-377*

- 3) You state that serological data assures that parents are blinded to exposure status. What about the positive PCR-test results? Also, I would expect that the parents would know about any symptomatic infections among their children; at least they would suspect that the child had been infected. Please comment?

Response: We did not state that parents were blinded to exposure status. We only stated that “Parents were not aware of their child(ren) serological test results while answering the questionnaire” *Lines 180-181*.

We agree with the reviewer that parents of children were likely aware of the exposure status as some children may have had a confirmed COVID-19 diagnosis or suffered from a symptomatic infection. However, in view of the large increase in SARS-CoV-2 and common respiratory infections in children, as well as the lack of systematic testing of children at the time of infection, it is likely that some SARS-CoV-2 infections were not detected. Using serological data helps decrease the bias of overestimating post-COVID when knowing the child had been infected, taking into consideration individuals who knew their child(ren) were infected and those who did not know.

We adapted the discussion with the following adjustments.

“Despite the fact that some parents were probably aware of their children’s exposure status when answering the questionnaire due to confirmed COVID-19 diagnosis or a symptomatic infection, relying on serological tests mitigates the bias of parents over-reporting persistent symptoms when knowing their child(ren) had been infected [35].” *Lines 365-368*

[35] Matta J, Wiernik E, Robineau O, Carrat F, Touvier M, Severi G, et al. Association of Self-reported COVID-19 Infection and SARS-CoV-2 Serology Test Results With Persistent Physical Symptoms Among French Adults During the COVID-19 Pandemic. *JAMA Intern Med.* 2022;182(1):19–25

- 4) Participation rate was on the low side (20%). Please comment?

Response: Recruiting is known as a highly challenging task in pediatric population-based cohort studies. A participation rate around 20% is common in children surveys. In this study, we faced the additional difficulty that parents were often reluctant to have their children, especially young ones, undergo a blood draw for research purposes only. We also noticed that some parents were tired of hearing about the pandemic, which also represented a strong barrier to recruitment. We have now detailed the participation rate in the discussion as follows:

“Despite the high number of recruited children from a wide age range, the participation rate was relatively low, although in line with participations rates in children cohort in other countries [36]. In addition to the general challenge of recruiting children in population-based cohort [37], we faced the additional difficulty that parents were often reluctant to have their children, especially young ones, undergo a blood draw for research purposes only. We also noticed that some parents were fed up or tired of hearing about the pandemic, which also represented a strong barrier to recruitment.” *Lines 382-388*

[36] Renk H, Dulovic A, Seidel A, Becker M, Fabricius D, Zernickel M, et al. Robust and durable serological response following pediatric SARS-CoV-2 infection. *Nat Commun.* 2022;13(1):1–11

[37] Barbieri V, Wiedermann CJ, Kaman A, Erhart M, Piccoliori G, Plagg B, et al. Quality of Life and Mental Health in Children and Adolescents after the First Year of the COVID-19 Pandemic: A Large Population-Based Survey in South Tyrol, Italy. *Int J Environ Res Public Health.* 2022;19(9):5220.

- 5) Lines 208-209: Please provide more details on what is meant by “marginal prediction after logistic regression”.

Response: We added the following details to clarify this approach.

“We used marginal prediction after logistic regression to estimate prevalence and prevalence difference, adjusting for age and sex. This statistical approach corresponds to models in which conditional predicted probabilities are calculated for each exposure level with every confounder fixed at its mean value.” *Lines 231-233*

- 6) Table 1: Please clarify if “Confirmed SARS-CoV-2 symptomatic infection” is based on the parental recall information that is mentioned in lines 177-179.

Response: Clarifications were added in Table 1.

- 7) Why do you only adjust for age and sex, and not more of the variables from table 1 such as chronic condition, financial situation and parental education.

Response: We thank the reviewer for this relevant input. We also conducted additional analysis adjusting for additional covariates such as chronic condition, financial situation and parental education. Since differences between different adjustments were minor, we decided to present the most parsimonious model. The following detail was added in Table 2.

“A model additionally adjusting for chronic condition, financial situation and parental education yielded similar results”

- 8) Please provide 1) the median number of symptoms reported among test-positives and test-negatives, respectively, 2) the median number of symptoms lasting over 12 weeks among test-positives and test-negatives, respectively.

Response: Thank you for this valuable comment. As suggested, the median numbers of symptoms lasting over 4 weeks and 12 weeks were added in Table 1.

- 9) How is the mean of the severity score calculated; I assume, that each child can contribute with more than one persistent symptom.

Response: We thank the reviewer for raising that point. Each child could experience multiple persistent symptoms and the mean severity score was evaluated considering the most severe symptoms of the symptomatic episode. The sentence was clarified as followed.

“The severity of persistent symptoms was then evaluated by asking parents “considering the most severe symptom of the episode, to what extent did this symptom affect the child's

daily life (on a scale of 1 very weak limitation - 10 strong limitation)” (On a scale from 1: very low limitation to 10: strong limitation).” *Lines 184-188*

- 10) Please consider presenting the mean of the severity score by specific symptom and by sero-pos./sero.neg.

Response: We agree with the reviewer that it would be interesting to present the average severity per symptoms. However, the limited sample size when stratifying per symptoms and serological status precluded us from conducting subgroup analyses.

- 11) Table 2: Please also provide prev.diff. according to sex.

Response: We thank the reviewer for this relevant proposition. The prevalence difference (Δ prev) between seronegative and seropositive in females was of 3.4% (95%CI: 1.1;8.4), and 4.7% (95%CI: 0.2;9.4) in males. The adjusted prevalence differences were presented in the result section and added in Table 2.

“After stratification by sex, the prevalence in females was of 3.4% (95%CI: 1.1;8.4) and in males, of 4.7% (95%CI: 0.2;9.4).” *Lines 251-253*

- 12) Table 3: Why not present these associations within test-negatives instead of within all? It is not clear to me that these “risk factors” are unique to test-positives. Why not estimate a ratio of ratios between test-positives and test-negatives to see if there is a difference- .

Response: We thank the reviewer for this interesting remark. First, we added in Table 3 the prevalence ratio of being seropositive and experiencing persistent symptoms for more than 12 weeks. Seropositives were almost twice as likely to experience symptoms lasting more than 12 weeks. We conducted a subgroup analysis within seropositives in order to identify further risk factors of post-COVID condition. As highlighted by the reviewer, no difference in prevalence ratio was observed between seronegatives and seropositives, even without computing ratio of ratios but only looking at the 95% confidence interval. This suggests that, in general, suffering from persistent symptoms for more than 12 weeks is more frequent among children with more disadvantaged socio-economical conditions or having a chronic condition, and it is exacerbated by having had a SARS-CoV-2 infection. We added in Table 3 the prevalence ratio of the serological status and the analysis within seronegatives. We also modified the following sentence.

“Older age (aPR 1.1, 95%CI: 1.0;1.2, continuous variable), being seropositive (aPR 1.8, 95%CI: 1.2;2.8), suffering from a chronic condition (aPR 3.6, 95%CI: 2.3;5.5) and living in a household with an average to poor financial situation (aPR 2.5, 95%CI: 1.4;4.6) were risk factors for experiencing persistent symptoms.”*Lines 276-279*

- 13) Table S1: Please replace “ans” in the column headings.

Response: We thank the reviewer for noticing this error. The table was corrected accordingly.

REVIEWERS' COMMENTS

Reviewer #2 (Remarks to the Author):

The authors have provided responses to all authors, including myself. I am satisfied with current version.

I am still unsure about response 11 to my comment (reviewer 2), and I would still mention that point as a possible limitation.

The new reference from Heiss et al, appropriately included, is probably published in the final version now.

My only concern is the 20% response rate, but I leave the author according to the journal standard to evaluate this point.

Reviewer #3 (Remarks to the Author):

Thank you for carefully addressing my comments. I have no further comments. Congratulations on providing a valuable contribution to the issue of long-covid in children and adolescents.

Second round of review 25.10.2022
Point-by-point responses

Page numbers refer to the version of the manuscript with revisions.

Comments from Reviewer #2

1. The authors have provided responses to all authors, including myself. I am satisfied with current version.

We thank the reviewer for this remark and for the previous comments that helped improving our manuscript.

2. I am still unsure about response 11 to my comment (reviewer 2), and I would still mention that point as a possible limitation.

In line with this comment, we added the following sentence in the discussion:

“The lack of difference in reported persistent symptoms in children younger than 12 years could be partly explained by the relatively small sample size and by symptoms often developed after common respiratory infections other than SARS-CoV-2 in this age range.”
Page 9, Lines 188-190

3. The new reference from Heiss et al, appropriately included, is probably published in the final version now.

We updated the manuscript with the recent reference.

Heiss R, Tan L, Schmidt S, Regensburger AP, Ewert F, Mammadova D, et al. Pulmonary Dysfunction after Pediatric COVID-19. *Radiology*. 2022;221250.

4. My only concern is the 20% response rate, but I leave the author according to the journal standard to evaluate this point.

We thank the reviewer for this remark, we rephrased the limitations as follows:

“Despite the high number of randomly recruited children from a wide age range, the participation rate was relatively low, although in line with participations rates in children

cohort in other countries [25]. In addition to the general challenge of recruiting children in population-based cohort [26], we faced the additional difficulty that parents were often reluctant to have their children, especially young ones, undergo a blood draw for research purposes only. We also noticed that some parents were tired of hearing about the pandemic, which also represented a strong barrier to recruitment. Finally, individuals with favourable socio-economic conditions were more likely to participate. This could have led to an underestimation of the prevalence of post-COVID since their occurrence was more common among underprivileged individuals. Overall, this might limit the representativeness of our results.” *Page 12, Line 259-262*

Comments from Reviewer #3

Thank you for carefully addressing my comments. I have no further comments. Congratulations on providing a valuable contribution to the issue of long-covid in children and adolescents. We thank a lot the reviewer for this statement.